

# The relationship between low energy availability, injuries, and bone health in recreational female athletes

Marta Gimunová[1], Michal Bozděch[2], Martina Bernaciková[1],
Romulo Fernandes[3], Michal Kumstát[4] and Ana Paludo[4]

[1] Department of Physical Activities and Health Sciences, Faculty of Sports Studies, Masaryk University, Brno, Czech Republic
[2] Department of Physical Education and Social Sciences, Faculty of Sports Studies, Masaryk University, Brno, Czech Republic
[3] Laboratory of InVestigation in Exercise–LIVE, Department of Physical Education, Sao Paulo State University (UNESP), Sao Paulo, Brazil
[4] Department of Sport Performance and Exercise testing, Faculty of Sports Studies, Masaryk University, Brno, Czech Republic

Corresponding author
Marta Gimunová,
gimunova@fsps.muni.cz

## ABSTRACT

**Background:** Low energy availability (LEA) causes pathophysiology of the female athlete's body affecting the bone and reproductive health and was observed to have a high prevalence in recreational female athletes previously. The aim of this study was to analyse the relationship between low energy availability in females questionnaire (LEAF-Q), bone mineral density (BMD), and postural stability in recreational athletes.

**Methods:** Recreational female athletes ($n$ = 24, age: 23.71 ± 2.94, Tier I) completed LEAF-Q, postural stability measurement during quiet stance (Zebris platform FDM; GmbH) and their BMD was measured using DXA (Hologic QDR Horizon A). Non-parametric statistical tests were used to analyse the relationships between LEAF-Q, BMD, and postural stability and to compare differences between participants divided by the LEAF-Q score and its subscales.

**Results:** Risk of LEA was observed in 50% of recreational athletes participating in this study. Up to 46% of participants perceived menstrual bleeding changes related to training and 37.50% experienced menstrual dysfunction. Body composition and body weight fluctuations were observed to affect postural stability and BMD. With the risk score for LEA, the BMD and postural stability were not negatively affected in recreational athletes. However, the high number of recreational athletes in the risk score for LEA and menstrual dysfunctions highlights the need for public health programs aimed to increase awareness of LEA and its health consequences and for open communication about the menstrual cycle. Future longitudinal studies observing LEA, BMD, menstrual function, postural stability, and their interrelationship in female athletes are needed to increase the knowledge of this topic.

## INTRODUCTION

Low energy availability (LEA)—an imbalance between energy intake and energy expenditure during physical activity—causes a pathophysiological state of the female athlete's body. LEA has a negative impact on several systems of the body, such as disruption of hypothalamic-pituitary-gonadal axis, absence of menstrual bleeding and the lack of oestrogen negatively affects bone health contributing to a decreased bone mineral density and subsequent increased risk of stress fractures (*Mountjoy et al., 2018*). Metabolic rate, manifested by decreasing resting metabolic rate, and exercise recovery are also negatively impacted by LEA (*Nattiv et al., 2007*; *Mountjoy et al., 2018*).

These health impairments and their mechanisms were described as a Female Athlete Triad (Triad) (*Nattiv et al., 2007*) and/or as a part of a newer concept of relative energy deficiency in sport (RED-S) (*Mountjoy et al., 2018*). The potentially irreversible health impairments highlight the need for prevention, early diagnosis, and treatment of female athletes (*Nattiv et al., 2007*). In elite female athletes, the elevated risk of the Triad and RED-S has been documented (*Logue et al., 2020*), while the issue is less investigated in recreational female athletes. The few data available reports that the prevalence of LEA in recreational female athletes might range from 35% to 69% (*Meng et al., 2020*; *Slater et al., 2016*; *Torstveit & Sundgot-Borgen, 2005*; *Sharps et al., 2022*).

Another aspect less explored among recreational athletes is the relationship between body composition (*e.g.*, bone, adipose tissue, and muscle mass), postural stability, and injuries. Bone mineral density (BMD) seems related to postural stability in a complex interaction that might affect the risk of fractures (*Simon et al., 2021*) due to the relevance of balance on injury prevention in sports (*Verhagen et al., 2004*; *McGuine & Keene, 2006*). Moreover, body weight fluctuation seems to affect postural control as well (*Fontana et al., 2009*), while postural control is a complex task maintaining the centre of mass within the base of support during a quiet stance and its impairment might be caused by musculoskeletal, visual, vestibular, somatosensory, or central nervous system disorders (*Horak, 2006*).

Most of the aspects discussed through this section were investigated in professional athletes, while the same phenomenon is less investigated among recreational athletes, especially among women. Considering the number of recreational athletes is expressively higher than professional athletes, its relevance in terms of population level cannot be ignored.

Therefore, the aim of this study was to analyse the relationship between LEAF-Q, injuries and bone health in recreational female athletes. It was hypothesized that in recreational athletes, the risk score for LEA will affect the body composition, BMD and postural stability. Additionally, it was hypothesized that menstrual (dys) functions and previous injuries will be related to BMD and body composition alterations.

## MATERIALS AND METHODS

### Participants

The study has been publicized in platforms of social media targeting university students in the metropolitan area of Brno, Czech Republic. Potential participants who contacted the research team had their inclusion criteria checked, in this case: female gender, classification of physical activity as Tier 1: recreationally active (according to the characteristics by *McKay et al., 2022*), age between 18 and 35 years, nulliparous, no diagnosed chronic illness, not pregnant, no medication use (including contraceptives) at the time of data collection, no serious injury on lower limbs and no diagnosed balance problems. For those participants who successfully met all inclusion criteria, researchers asked to sign a written informed consent form (mandatory to take part of the study). The study was conducted in accordance with Declaration of Helsinki and was approved by the Research Ethics Committee of Masaryk University (EKV-2021-109). Data collection was performed from April to June 2022, while a group of 24 Czech recreational female athletes (Caucasian) participated in this study.

In Fig. 1, this study design flow chart is shown.

### LEAF-Q

Participants completed LEAF-Q which includes questions related to the training, gastrointestinal function, injuries during the previous year, and menstrual function since menarche. The LEAF-Q score of eight or higher suggests a risk for the LEA (*Melin et al., 2014*). Furthermore, questions from specific subscales of LEAF-Q were used separately in the statistical analysis. Bone health subscale included question on the occurrence of injuries in the past year in this study (*Have you had absences from your training, or participation in competitions during the last year due to injuries?*). In this study, the menstrual function subscale included questions about the age of menarche (*How old were you when you had your first period?*), menstrual bleeding changes related to training (*Do you experience that your menstruation changes when you increase your exercise intensity, frequency or duration?*), and the history of menstrual dysfunctions (*Have your periods ever stopped for three consecutive months or longer (besides pregnancy)?*).

The low energy availability in females questionnaire (LEAF-Q) promotes the early detection of LEA in female athletes considering training, gastrointestinal function, injuries during the previous year, and menstrual function since menarche. The LEAF-Q score of 8 or higher suggests a risk for the Triad and RED-S (*Melin et al., 2014*). Previous study suggests that LEAF-Q is a practical tool to identify LEA also in recreational female athletes identifying those in risk of impaired bone health (*Black et al., 2018*).

### Body composition and anthropometry

Body composition (total body mass (kg), fat free mass (kg), fat (%), and BMD (g/cm$^2$)) were measured using dual-energy X-ray absorptiometry (Hologic QDR Horizon A). Participants were scanned according to the manufacturer's standard protocol. As the variability in BMD among skeletal sites exists, the BMD in the femoral neck, lumbar spine, and whole body was measured as recommended by *Nattiv et al. (2007)*. A low BMD is

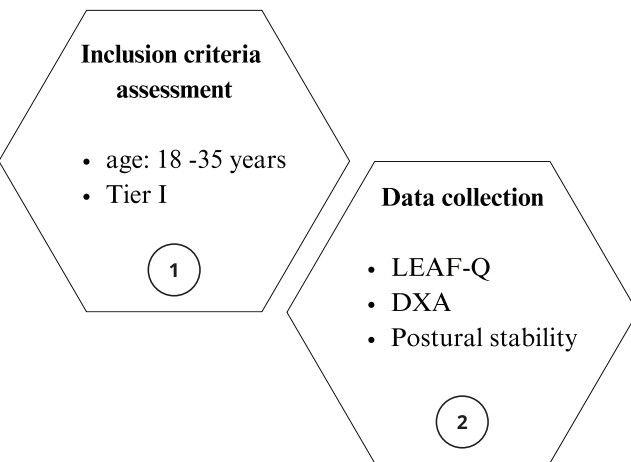

**Figure 1 Study design flow chart.**

defined as Z-score lower than −2.0 in young adults (*Mäkitie & Zillikens, 2022*). In athletes, the American College of Sports Medicine defines low BMD as Z-score between −1.0 and −2.0 along with LEA, amenorrhea, or a history of stress fractures (*Nattiv et al., 2007*). Body height was measured by a stadiometer (Kern, MPE, Kern and SOHN, GmbH, Germany). Body weight fluctuation (kg) was counted as the difference between the highest and lowest body weight at the current body height.

## Postural stability

Postural stability was assessed at two conditions: with eyes opened (EO) and eyes closed (EC) as used in previous studies (*e.g.*, *Andreeva et al., 2021*; *Gimunová et al., 2020*). Participants were asked to stand still as possible on the Zebris platform (FDM; GmbH, Munich, Germany) with a narrow stance (heels and big toes touching) for 30 s. From the Zebris software, the average CoP velocity (mm/s) was obtained for both conditions. Lower CoP velocity has been associated with increased postural stability in previous studies (*e.g.*, *Andreeva et al., 2021*; *Gimunová et al., 2020*).

## Statistical analysis

Most of the variables did not meet the assumptions of normal distribution tested by the Shapiro–Wilk test. The Mann–Whitney U test was used to compare the differences for body composition, BMD and postural stability variables according to LEAF-Q score (Risk Group (score ≥ 8) and Non-Risk Group (score < 8)), bone health subscale (injuries in the past year (Yes or No)) and menstrual function subscale (age of menarche (≤15 years-old and >15 years-old)), perceived menstrual bleeding changes related to training (Yes or No), and the history of menstrual dysfunctions (oligomenorrhea/amenorrhea: Yes or eumenorrheic: No).

The relationship between LEAF-Q, body composition, BMD and postural stability was tested by Spearman's Rho. SPSS Statistics (IBM SPSS Statistics for Windows, Version 28.0; IBM Corp., Armonk, NY, USA) was used for the statistical analysis. Significance level (*p*-value) was set at <5% (statistically significant).

To assess the robustness of this study's test, a power analysis was conducted. The power analysis allowed evaluation of the likelihood of detecting an effect, if present, based on the sample size, using the LEAF-Q score as the dependent variable and changes in menstrual bleeding related to training (Yes or No) as the independent variable. The analysis revealed a high probability of detecting an effect, with a significant value of 1-beta = 0.964.

The *a priori* sample size was calculated with an alpha level set at 0.05 based on the "perceived menstrual bleeding changes related to training" which is a part of LEAF-Q (*Melin et al., 2014*) and was chosen as it might be one of the first indicators of altered reproductive functions due to LEA. The sample size calculation showed that the minimum sample size required in each group was eight for the "Yes group" and six for the "No group" for "perceived menstrual bleeding changes related to training", ensuring the desired probability of effect detection. With the current sample size of 13 and 11 participants in "perceived menstrual bleeding changes related to training," Yes or No groups respectively, the power (1-beta) rises to 0.9988.

## RESULTS

### LEAF-Q score

Participants' characteristics divided by LEAF-Q score and differences between the risk and non-risk groups for LEA are shown in Table 1. The LEAF-Q score in 12 of the study participants (50%) suggested the risk of LEA. A Z-score range from −1.0 to 1.6 in BMD total body measurement was observed. The Z-score of −1.0 was observed in two recreational female athletes (8.33%; risk group: $n = 1$, non-risk group: $n = 1$) in this study. As shown in Table 1, the statistically significant difference between risk and non-risk groups of LEAF-Q score was observed in femoral neck BMD. The BMD was higher in the risk LEAF-Q group of recreational athletes.

The occurrence of injury during the past year did not significantly affect the body composition, BMD nor postural stability in recreational female athletes as shown in Table 2. However, the occurrence of injury in the past year led to a significantly higher LEAF-Q score.

The age of menarche did not significantly affect the body composition, BMD nor postural stability in recreational female athletes. The presence of menstrual dysfunction (oligomenorrhea or secondary amenorrhea) led to a significantly higher body weight fluctuation. In recreational athletes who perceived menstrual bleeding changes related to training significantly lower LEAF-Q score, higher body mass and higher femoral neck BMD were observed (Table 3).

The results of Spearman's Rho showing the relationship between LEAF-Q, body composition, BMD and postural stability are shown in Table 4. A significant correlation of BMD was observed with the total body mass ($r = 0.551$) and fat free mass ($r = 0.890$). When having eyes open no correlation between LEAF-Q score or BMD and postural stability variables was observed. Postural stability in eyes closed condition showed a significant negative correlation with body weight fluctuations ($r = −0.409$), current BMI ($r = −0.543$) and total body mass ($r = −0.452$).

**Table 1 General characteristics of the participants divided by the LEAF-Q score (n = 24).**

| Variables | Low energy availability | | | | U | p |
|---|---|---|---|---|---|---|
| | Risk Group (n = 12) | | Non-Risk Group (n = 12) | | | |
| | Med | (x25–x75) | Med | (x25–x75) | | |
| LEAF-Q score | 10.00 | (9.00–11.00) | 1.50 | (1.00–2.00) | 0.0 | 0.001* |
| Age (years) | 23.00 | (22.00–24.00) | 23.00 | (21.75–57.75) | 71.5 | 0.977 |
| Body mass (kg) | 65.70 | (61.27–67.97) | 58.92 | (55.45–68.37) | 52.0 | 0.266 |
| Body height (cm) | 166.00 | (164.75–172.00) | 163.25 | (610.00–170.50) | 53.0 | 0.291 |
| BMI | 22.27 | (21.28–23.95) | 21.98 | (20.42–24.19) | 65.0 | 0.713 |
| Body weight fluctuation (kg) | 14.00 | (8.50–16.00) | 9.00 | (7.00–14.00) | 50.5 | 0.219 |
| Z-score total body | 0.55 | (−0.03–1.20) | 0.05 | (−0.28–0.42) | 52.5 | 0.266 |
| BMD total body (g/cm$^2$) | 1.14 | (1.10–1.18) | 1.11 | (1.8–1.14) | 56.0 | 0.378 |
| BMD L spine (g/cm$^2$) | 1.09 | (1.05–1.20) | 1.06 | (1.03–1.10) | 49.0 | 0.198 |
| BMD neck (g/cm$^2$) | 1.01 | (0.94–1.01) | 0.88 | (0.81–0.93) | 35.0 | 0.033* |
| Fat free mass (kg) | 45.63 | (43.46–47.98) | 42.28 | (38.50–46.52) | 50.0 | 0.219 |
| Fat mass (%) | 27.15 | (26.37–31.15) | 28.80 | (27.95–33.75) | 98.0 | 0.143 |
| EO CoP average velocity (mm/s) | 8.54 | (7.46–10.20) | 10.21 | (8.74–11.90) | 101.0 | 0.101 |
| EC CoP average velocity (mm/s) | 16.38 | (14.48–17.63) | 18.22 | (16.32–24.50) | 97.0 | 0.160 |

Notes:
Mann–Whitney U test; LEA, low energy availability; BMI, body mass index; BMD, bone mineral density; EO, eyes opened; EC, eyes closed.
* Highlights the statistical significance.

**Table 2 Comparison between groups based on the bone health subscale.**

| | Injury in the past year | | | | U | p |
|---|---|---|---|---|---|---|
| | No (n = 15) | | Yes (n = 9) | | | |
| | Med | (x25–x75) | Med | (x25–x75) | | |
| LEAF-Q score | 2.00 | (1.00–2.50) | 10.00 | (9.00–11.00) | 14.0 | 0.001* |
| Body mass (kg) | 60.36 | (55.73–70.40) | 64.73 | (61.60–67.81) | 55.0 | 0.482 |
| Body weight fluctuation (kg) | 10.00 | (7.25–14.00) | 15.00 | (8.00–16.00) | 55.5 | 0.482 |
| BMD total body (g/cm$^2$) | 1.11 | (1.07–1.18) | 1.13 | (1.10–1.17) | 58.0 | 0.599 |
| BMD L spine (g/cm$^2$) | 1.08 | (1.04–1.12) | 1.08 | (1.05–1.13) | 62.0 | 0.770 |
| BMD neck (g/cm$^2$) | 0.89 | (0.80–1.01) | 1.00 | (0.95–1.04) | 36.0 | 0.064 |
| Fat free mass (kg) | 43.29 | (38.72–48.45) | 44.90 | (44.10–47.66) | 54.0 | 0.446 |
| Fat mass (%) | 28.60 | (27.40–31.60) | 27.10 | (26.40–33.10) | 85.0 | 0.318 |
| EO CoP average velocity (mm/s) | 9.90 | (7.51–11.02) | 8.79 | (7.63–10.17) | 85.0 | 0.318 |
| EC CoP average velocity (mm/s) | 16.98 | (14.31–22.69) | 16.56 | (15.88–18.57) | 72.0 | 0.815 |

Notes:
Mann–Whitney U test; BMD, bone mineral density; EO, eyes opened; EC, eyes closed.
* Highlights the statistical significance.

In Fig. 2, the statistically significant correlations and results from differences between groups were summarised to present an overview of interrelationships of LEAF-Q score, BMD of femoral neck, total body mass and CoP average velocity at EC condition.

Table 3 Comparison between groups based on the menstrual function subscale.

| | | Age of menarche[A] | | Menstrual bleeding changes related to training | | Menstrual dysfunction | |
|---|---|---|---|---|---|---|---|
| | | Up to 15 (n = 19) | 15+ (n = 4) | Yes (n = 11) | No (n = 13) | Yes (n = 9) | No (n = 15) |
| LEAF-Q score | Med | 3.0 | 6.00 | 10.00 | 2 | 9.00 | 2.00 |
| | (x25–x75) | (1.15–99.50) | (1.75–10.00) | (8.50–10.50) | (1.00–2.00) | (2.00–11.00) | (1.00–9.50) |
| | U | 36.0 | | 16.5 | | 38.5 | |
| | p | 0.907 | | 0.001* | | 0.084 | |
| Body weight fluctuation (kg) | Med | 14.00 | 6.75 | 13.00 | 10.00 | 16.00 | 8.00 |
| | (x25–x75) | (8.00–16.00) | (6.00–8.63) | (9.00–16.00) | (7.00–14.00) | (10.00–16.00) | (7.00–14.00) |
| | U | 63.0 | | 52.5 | | 33.5 | |
| | p | 0.054 | | 0.277 | | 0.041* | |
| BMD total body (g/cm$^2$) | Med | 1.12 | 1.14 | 1.15 | 1.11 | 1.13 | 1.11 |
| | (x25–x75) | (1.09–1.17) | (1.10–1.18) | (1.10–1.18) | (1.08–1.13) | (1.08–1.15) | (1.10–1.18) |
| | U | 36.0 | | 60.0 | | 79.0 | |
| | p | 0.785 | | 0.531 | | 0.519 | |
| BMD L spine (g/cm$^2$) | Med | 1.08 | 1.13 | 1.09 | 1.05 | 1.05 | 1.09 |
| | (x25–x75) | (1.04–1.12) | (1.03–1.23) | (1.05–1.17) | (1.04–1.12) | (1.05–1.09) | (1.04–1.13) |
| | U | 30.5 | | 55.0 | | 76.5 | |
| | p | 0.611 | | 0.361 | | 0.599 | |
| BMD neck (g/cm$^2$) | Med | 0.95 | 0.99 | 1.02 | 0.89 | 0.89 | 0.95 |
| | (x25–x75) | (0.84–1.02) | (0.90–1.07) | (0.97–1.06) | (0.82–0.90) | (0.82–1.04) | (0.87–1.02) |
| | U | 27.0 | | 34.0 | | 73.0 | |
| | p | 0.324 | | 0.030* | | 0.77 | |
| Fat free mass (kg) | Med | 44.90 | 44.82 | 46.36 | 41.27 | 46.53 | 44.79 |
| | (x25–x75) | (39.28–49.74) | (42.78–46.68) | (44.45–48.31) | (38.65–45.16) | (40.59–51.48) | (40.03–45.79) |
| | U | 38.0 | | 41.0 | | 51.0 | |
| | p | 0.785 | | 0.082 | | 0.347 | |
| Body mass (kg) | Med | 62.48 | 62.55 | 66.83 | 57.48 | 66.83 | 62.28 |
| | (x25–x75) | (56.60–58.97) | (59.47–65.22) | (63.60–68.97) | (55.38–62.28) | (55.76–69.51) | (57.14–66.56) |
| | U | 43.0 | | 33.0 | | 57.0 | |
| | p | 0.969 | | 0.026* | | 0.558 | |
| Fat mass (%) | Med | 28.60 | 27.80 | 27.20 | 28.30 | 28.10 | 29.00 |
| | (x25–x75) | (26.95–33.35) | (27.08–28.85) | (26.35–33.35) | (27.50–30.30) | (26.80–28.30) | (27.20–34.30) |
| | U | 45.0 | | 81.0 | | 89.0 | |
| | p | 0.785 | | 0.608 | | 0.215 | |
| EO CoP average velocity (mm/s) | Med | 9.90 | 8.32 | 8.29 | 9.90 | 9.46 | 9.04 |
| | (x25–x75) | (7.59–10.56) | (7.68–8.96) | (7.37–10.39) | (8.96–11.25) | (7.63–10.32) | (7.70–10.93) |
| | U | 50.0 | | 95.0 | | 76.0 | |
| | p | 0.324 | | 0.186 | | 0.640 | |

(Continued)

| Table 3 (continued) | | Age of menarche[A] | | Menstrual bleeding changes related to training | | Menstrual dysfunction | |
|---|---|---|---|---|---|---|---|
| | | Up to 15 (*n* = 19) | 15+ (*n* = 4) | Yes (*n* = 11) | No (*n* = 13) | Yes (*n* = 9) | No (*n* = 15) |
| EC CoP average velocity (mm/s) | Med | 16.98 | 17.50 | 16.20 | 19.37 | 16.56 | 18.57 |
| | (x25–x75) | (15.46–22.32) | (15.02–20.23) | (13.62–16.96) | (16.37–23.56) | (14.78–17.32) | (16.04–23.62) |
| | U | 40.0 | | 105.0 | | 91.0 | |
| | p | 0.907 | | 0.055 | | 0.174 | |

Notes:
Mann–Whitney U test; Mann–Whitney U test; BMD, bone mineral density; EO, eyes opened; EC, eyes closed.
[A] One participant who did not remember the age of menarche was not included in the analysis.
* Highlights the statistical significance.

The overview shows a higher number of interrelationships of BMD, body mass and CoP average velocity than with a LEAF-Q score.

# DISCUSSION

This cross-sectional study aimed to analyse the complex relationship between LEAF-Q, injuries, menstrual cycle health, postural stability and bone health in recreational female athletes. The significant interrelationships between analyzed parameters show a novel field for research and possible prevention programs focused on decreasing the risk of falls in LEA and/or low BMD populations. Body composition and body weight fluctuations were observed to affect postural stability and BMD in this study. The risk of LEA was observed in 50% of recreational athletes. Up to 46% of participants perceived menstrual bleeding changes related to training and 37.50% experienced menstrual dysfunction. As recreational female athletes are underrepresented in current research (*Slater et al., 2016*), this study brings valuable insight into this population.

Recreational female athletes seem to be at increased risk of LEA compared to elite athletes (*Torstveit & Sundgot-Borgen, 2005*). This phenomenon seems partially supported by the fact that recreational athletes have less access to the nutritional and sport medicine specialists (*Slater et al., 2016*) and highlights the need for public health programs raising the awareness of LEA and its health consequences among this specific group. Every female athlete should be aware about energy intake to compensate for energy expenditure to preserve eumenorrhea and bone health. Despite the low number of participants, the observed high prevalence of athletes at the risk of LEA (50%) from the Czech Republic in this study is in accordance with the worldwide prevalence of LEA in recreational athletes ranging from 35% to 69% (*Meng et al., 2020*; *Slater et al., 2016*; *Torstveit & Sundgot-Borgen, 2005*; *Black et al., 2018*; *Sharps et al., 2022*).

The aim of this study was to assess the relationship between the LEAF-Q, body composition, BMD and postural stability in recreational athletes (Tier I as defined in *McKay et al., 2022*). Differences in the terminology of recreational athlete in previous studies may result in contradictory observations. Previous study confirmed LEA determined by diet records and exercise diaries, low calcium intake and reductions in T3,
**Table 4 Correlation matrix of the relationship between LEAF-Q, BMD, and postural stability.**

| Variable | | LEAF-Q | Body weight fluctuations (kg) | Body height (cm) | BMI | BMD total body (g/cm²) | BMD L spine (g/cm²) | BMD neck (g/cm²) | Fat free mass (kg) | Body mass (kg) | Fat mass (%) | Training hours/week | EO COP average velocity, mm/sec | EC COP average velocity, mm/sec |
|---|---|---|---|---|---|---|---|---|---|---|---|---|---|---|
| Body weight fluctuations (kg) | rho | 0.29 | | | | | | | | | | | | |
| | $p$ | 0.16 | | | | | | | | | | | | |
| Body height (cm) | rho | 0.14 | 0.09 | | | | | | | | | | | |
| | $p$ | 0.53 | 0.66 | | | | | | | | | | | |
| BMI | rho | 0.10 | 0.47 | −0.25 | | | | | | | | | | |
| | $p$ | 0.64 | 0.02* | 0.23 | | | | | | | | | | |
| BMD total body (g/cm2) | rho | 0.20 | 0.23 | 0.36 | 0.25 | | | | | | | | | |
| | $p$ | 0.36 | 0.28 | 0.09 | 0.23 | | | | | | | | | |
| BMD L spine (g/cm2) | rho | 0.29 | 0.05 | 0.04 | 0.17 | 0.72 | | | | | | | | |
| | $p$ | 0.16 | 0.82 | 0.85 | 0.44 | <0.001* | | | | | | | | |
| BMD neck (g/cm2) | rho | 0.34 | 0.13 | 0.32 | 0.45 | 0.80 | 0.57 | | | | | | | |
| | $p$ | 0.11 | 0.55 | 0.12 | 0.03* | <0.001** | $3.65 \times 10^{-3**}$ | | | | | | | |
| Fat free mass (kg) | rho | 0.24 | 0.40 | 0.45 | 0.58 | 0.46 | 0.15 | 0.61 | | | | | | |
| | $p$ | 0.26 | 0.05 | 0.03* | $3.51 \times 10^{-3**}$ | 0.03* | 0.49 | $1.97 \times 10^{-3**}$ | | | | | | |
| Body mass (kg) | rho | 0.21 | 0.52 | 0.30 | 0.77 | 0.38 | 0.08 | 0.55 | 0.89 | | | | | |
| | $p$ | 0.32 | $9.04 \times 10^{-3**}$ | 0.15 | <0.001** | 0.07 | 0.71 | $5.95 \times 10^{-3**}$ | <0.001** | | | | | |
| Fat mass (%) | rho | −0.29 | 0.19 | −0.32 | 0.37 | −0.26 | −0.18 | −0.27 | −0.26 | 0.11 | | | | |
| | $p$ | 0.17 | 0.38 | 0.13 | 0.08 | 0.21 | 0.39 | 0.20 | 0.22 | 0.60 | | | | |
| Training hours/week | rho | 0.50 | −0.08 | −0.07 | −0.03 | −0.01 | −0.02 | 0.33 | 0.23 | 0.03 | −0.53 | | | |
| | $p$ | 0.01* | 0.72 | 0.76 | 0.89 | 0.96 | 0.93 | 0.12 | 0.28 | 0.90 | $7.46 \times 10^{-3**}$ | | | |
| EO COP average velocity, mm/sec | rho | −0.18 | 0.06 | 0.15 | −0.39 | −0.05 | −0.06 | −0.34 | −0.15 | −0.27 | −0.17 | −0.17 | | |
| | $p$ | 0.40 | 0.78 | 0.48 | 0.06 | 0.82 | 0.79 | 0.10 | 0.49 | 0.20 | 0.43 | 0.42 | | |
| EC COP average velocity, mm/sec | rho | −0.35 | −0.41 | 0.12 | −0.54 | −0.18 | −0.14 | −0.30 | −0.26 | −0.45 | −0.33 | 0.05 | 0.58 | |
| | $p$ | 0.09 | 0.05* | 0.58 | $6.82 \times 10^{-3**}$ | 0.39 | 0.52 | 0.16 | 0.22 | 0.03* | 0.11 | 0.81 | $3.34 \times 10^{-3**}$ | |

**Notes:**

Spearman's Rho; WB, whole-body; LEA, low energy availability; BMI, body mass index; BMD, bone mineral density; EO, eyes opened; EC, eyes closed.

\* Correlation is significant at the 0.05 level (two-tailed).

\*\* Correlation is significant at the 0.01 level (two-tailed).

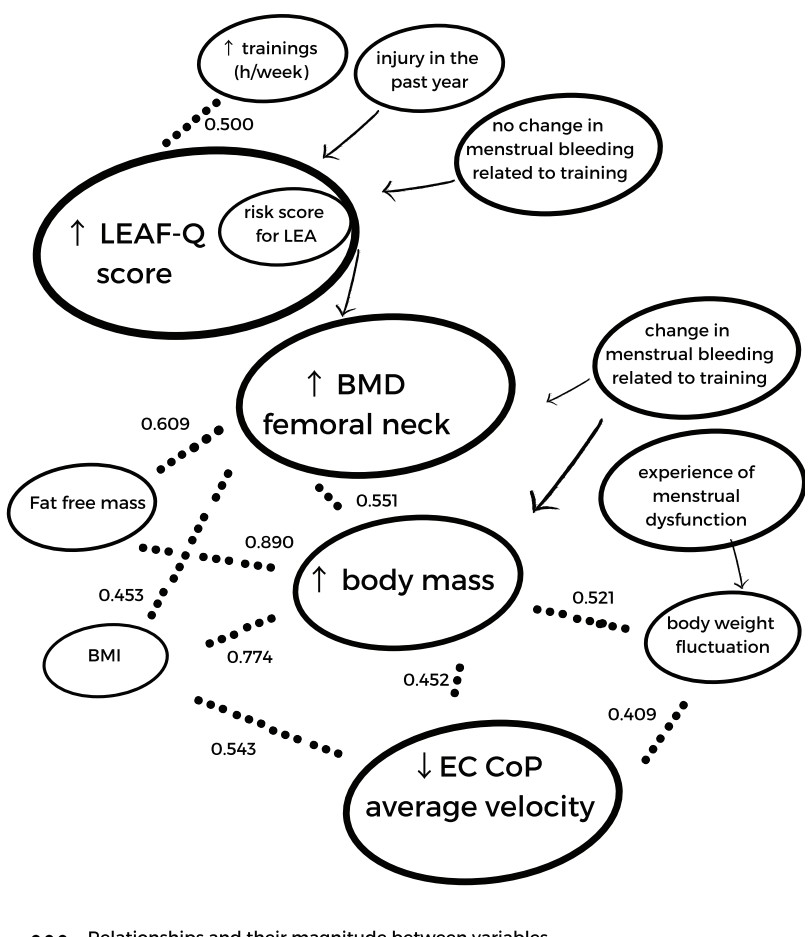

**●●●** Relationships and their magnitude between variables
⟶ Effects and their directions on main variables

**Figure 2** **The summary of statistically significant relationships (results of Spearman's Rho) and effects (results of Mann-Whitney U tests) between LEAF-Q score and its subscales, BMD of femoral neck, body mass and CoP average velocity at EC condition.** ↑ increasing value, ↓ decreasing value.

suggesting the increased risk of poor bone health in recreational athletes with risk score in LEAF-Q (*Black et al., 2018*). On high school female students, in both physically active and inactive girls, the components of Triad were observed, in which more inactive girls met the criteria of low BMD compared to active girls in that study (*Hoch et al., 2009*). In this study, a statistically higher BMD values were observed in the recreational athletes with risk score in LEAF-Q compared to non-risk group. As the moderate correlation between LEAF-Q score and trainings (h/week) was observed, this finding may be a result of the protective effect of weight-bearing physical activities on BMD as athletes from high-impact sports were observed to have significantly higher BMD compared to sedentary controls and/or low-impact sports (*Bellver et al., 2019*; *Kruusamäe, Maasalu & Jürimäe, 2016*; *Maillane-Vanegas et al., 2020*; *van Santen et al., 2019*).

Nevertheless, previous systematic review showed high heterogeneity in BMD among studies comparing elite female athletes and non-athletes, denoting both groups are prone to low BMD and LEA (*Skarakis et al., 2021*). In this study, none of the analysed groups

showed reduced BMD as defined by American College of Sports Medicine and the absence of differences between groups might be justified, at least in parts, by this background. Moreover, it is important to consider the current BMD (as well as its trend in time because BMD) reflects the athlete's cumulative history of energy availability, menstrual function, genetics, training routine and other factors (*Nattiv et al., 2007*) and, therefore, such phenomenon should be assessed under a perspective of longitudinal designed.

Good postural stability is essential not only for sports activities but also for activities of daily living. Musculoskeletal, proprioceptive, vestibular and visual systems are integrated to maintain the quiet standing (*Montesinos et al., 2018*; *Ołpińska-Lischka, Kujawa & Maciaszek, 2021*). Previous studies show that low BMD impairs the postural stability in patients with osteoporosis and/or in postmenopausal women (*Hita-Contreras et al., 2014*; *Simon et al., 2021*; *Burke et al., 2010*). As no participant in this study had low BMD (Z-score lower than −2.0 in young adults as defined in *Mäkitie & Zillikens, 2022*) no correlation of BMD and postural stability was observed in this study. In the best of our knowledge, there is no study assessing the effect of LEA/low BMD in athletes on postural stability despite the increased risk of fall and subsequent injury with postural instability (*Hita-Contreras et al., 2014*). In previous studies the anthropometric parameters and body weight fluctuations were observed to affect the postural stability (*Hue et al., 2007*; *Fontana et al., 2009*; *Ku et al., 2012*). Similarly, the total body mass, BMI, and body weight fluctuations were observed to affect the postural stability at EC condition in this study.

From menstrual status, menstrual bleeding changes related to training were observed to affect the BMD and total body mass in this study. More than a half of the elite female athletes (54.4%) reported changes in menstrual bleeding when increasing exercise intensity and volume in a previous study (*Majumder et al., 2022*). Similar percentage of recreational athletes in this study (45.83%) reported perceived change in menstrual cycle bleeding related to the increase of intensity or volume of training. Furthermore, 37.50% of participants in this study experienced menstrual dysfunction (oligomenorrhea or amenorrhea) and 17.40% reported the late onset of menarche (primary amenorrhea). The mean prevalence of menstrual dysfunction in elite athletes is up to 56% and primary amenorrhea has the mean prevalence up to 25% in some sports modalities (*Gimunová et al., 2022*). Still, only about 11% to 27% of elite athletes communicate about menstrual cycle with their coach (*Solli et al., 2020*; *von Rosen et al., 2022*) and most of the female athletes agrees that menstrual cycle is considered a taboo topic in sports (*von Rosen et al., 2022*; *Findlay et al., 2020*). Similarly, in European society there is a dearth of the open communication about menstrual cycle as observed (*Schoep et al., 2019*).

There are several limitations of this study. The low number of participants does not allow generalization of the results of this study and the cross-sectional design of this study does not allow stablishing causality pathways. The results of this study highlight the need for future studies on the prevalence of LEA among recreational athletes, especially longitudinal studies observing the trend of BMD and menstrual functions in time as data is lacking about the longitudinal effect of LEA in recreational athletes and its effect on BMD, menstrual function, and postural stability.

## CONCLUSIONS

LEA constitutes an event of high occurrence among recreational female athletes, which seems affected by training volume. Bone aspects and postural stability were outcomes not related to LEA, while the occurrence of injuries and reproductive aspects did. Although this study was the first, future longitudinal studies observing LEA, BMD, menstrual function, postural stability, and their interrelationship are needed for both recreational and elite female athletes, especially in sports modalities with increased demands on postural stability and balance.

### Funding

This work was supported by the Masaryk University under Grant MUNI/A/1389/2021. The funders had no role in study design, data collection and analysis, decision to publish, or preparation of the manuscript.

### Grant Disclosures

The following grant information was disclosed by the authors:
Masaryk University: MUNI/A/1389/2021.

### Competing Interests

The authors declare that they have no competing interests.

### Author Contributions

- Marta Gimunová conceived and designed the experiments, performed the experiments, prepared figures and/or tables, authored or reviewed drafts of the article, and approved the final draft.
- Michal Bozděch performed the experiments, analyzed the data, prepared figures and/or tables, authored or reviewed drafts of the article, and approved the final draft.
- Martina Bernaciková conceived and designed the experiments, authored or reviewed drafts of the article, and approved the final draft.
- Romulo Fernandes conceived and designed the experiments, authored or reviewed drafts of the article, and approved the final draft.
- Michal Kumstát conceived and designed the experiments, authored or reviewed drafts of the article, and approved the final draft.
- Ana Paludo conceived and designed the experiments, prepared figures and/or tables, authored or reviewed drafts of the article, and approved the final draft.

### Human Ethics

The following information was supplied relating to ethical approvals (*i.e.*, approving body and any reference numbers):

The study was approved by the Research Ethics Committee of Masaryk University (EKV-2021-109).

## Data Availability

The raw data is available in the Supplemental Files.

## Supplemental Information

Supplemental information for this article can be found online at http://dx.doi.org/10.7717/peerj.17533#supplemental-information.

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
