# Peer review of "The relationship between low energy availability, injuries, and bone health in recreational female athletes"

_PeerJ, doi:10.7717/peerj.17533_

## Round 0.1 · original submission · Major Revisions

The manuscript contains interesting data, but the reviewers have identified several areas that need revision.

·

Basic reporting

1. The manuscript is well-written and easy to understand
2. The background information is written in detail and the past related research works are also listed

Experimental design

1. The primary research aim and data collection are clearly defined and described.
2. The experiment materials and methods are well-defined.
3. The author mentioned most of the variables didn’t meet the assumptions of normal distribution in the statistical analysis part, and decided to use the Mann–Whitney U test, which makes sense. However, it would be great to include a statistical preliminary analysis part (such as showing data distribution by figure, etc.) before implementing the analysis.

Validity of the findings

1. Statistical results are well discussed and interpreted, and the conclusions are well-stated

·

Basic reporting

The study needs to be improved, especially in statistical terms. Additionally, the answers to my questions regarding the sample are very important for the validity of the study.

Experimental design

Summarize your methodology using a flow chart.

Also, how was the sample size determined? What was the realized power of the work? Under which testing procedure was the power analysis performed, such important information should be presented in the article for the validity of the findings.

Validity of the findings

Please provide relevant p values as well as r values in the results.


Give p and U values in the results for significance tests.

If the data is not normally distributed, why did you use mean and standard deviation?


For data that are not normally distributed, present the median and interquartile range as descriptive statistics.


The tests used should be stated below the table.

Additional comments

Results need to be significantly improved.

·

Basic reporting

Thank you for your submitted manuscript entitled, “The relationship between low energy availability, injuries, and bone health in recreational female athletes’’. The area of the research is interesting; however, it needs a few amendments. Overall, the paper is well-written. The manuscript is well-structured, and it is easy to follow the sections.

The introduction provides quite a proper background of the topic (12 references, but it could have been more if they are available) and there is also a nicely-presented novelty part.
The key words are different from the words in the title.
The quality of the images is good enough, but I don’t know if the reviewing version has lower resolution than the final version. If not, images should have better resolution in its final size.
It seems that the English is clear, but research articles usually do not use the word "we/our" and regularly use passive verbs (lines 78 and 79, as well as 203 and 212).

Experimental design

The experimental design meets the scope of the journal, and the hypotheses are well-defined and relevant to the community.
Was the number of the participants established by a priori power analyse?
Methods are described detailed enough. The four tables and Figure 1 illustrate the relationships presented in the study.

Validity of the findings

Most of the results are quite interesting and are well discussed.
In the Discussion it would be better to have seen more use of terms like 'originality' and 'significance' in the first paragraph. Identify what is new in this study that may benefit readers or how it may advance existing knowledge or create new knowledge on this subject. There should be a clear conclusion on why the research findings are significant (the limitations are well presented but the strengths somehow are overlooked).

---

## Round 0.2 · accepted · Accept

I confirmed that the authors have addressed all of the reviewers' comments and judged that this manuscript is worthy of publication.

·

Basic reporting

The article took all my comments into account in the second version. It is acceptable now.

Experimental design

Developed and suitable

Validity of the findings

In particular, statistical revisions have been significantly corrected.

·

Basic reporting

Thank you for providing this comprehensive work.
The authors have presented an improved version of the manuscript.

The introduction provides a proper background of the topic.
The sections have been improved. Relevant results are well-organized.
The quality of the images is good enough.

Experimental design

The experimental design meets the scope of the journal, and it is relevant to the community.
Methods are described detailed enough.

Validity of the findings

The results and the conclusions are quite interesting and well-discussed. All data are provided.

Additional comments

The authors have adequately addressed all my comments. I have no further suggestions.